# Numerical and Physical Simulation of MAG Welding of Large S235JRC+N Steel Industrial Furnace Wall Panel

**DOI:** 10.3390/ma16072779

**Published:** 2023-03-30

**Authors:** Marek Mróz, Robert Czech, Bogdan Kupiec, Andrzej Dec, Marcin Spólnik, Patryk Rąb

**Affiliations:** 1Department of Foundry and Welding, Faculty of Mechanical Engineering and Aeronautics, Rzeszow University of Technology, Al. Powstańców Warszawy 12, 35-959 Rzeszów, Poland; 2Naftoremont-Naftobudowa Sp. z o.o., Branch Jedlicze, ul. Sikorskiego 17, 38-460 Jedlicze, Poland

**Keywords:** MAG welding method, numerical and physical simulation, welding deformation, metallographic tests, hardness measurements

## Abstract

This paper presents the results of a study on the development of a Metal active gas (MAG) welding technology for an industrial furnace component made of steel S235JRC+N with respect to the minimizationof welding deformation. A numerical simulation of the welding process was performed in the first phase of the research. The numerical simulation was carried out with the SYSWELD software. For the numerical simulation of the welding process, the FEM method was used. In the simulation, four variants of restraint of the industrial furnace wall panel elements during the execution of the welding process were investigated. They differed in the number of restraints (model 1–4). It was found that the difference between the maximum mean strain in model 1 and the lowest mean strain in model 4 was only 11%. A physical simulation of the welding process was then performed with a restraint variant according to model 1. The displacement results obtained from the physical simulation of the welding process were compared with the displacement results from the numerical simulation. Discrepancies between numerical and physical simulation displacement values were found. The quality of selected welded joints was also evaluated. Visual testing (VT) and measurements of weld geometries were performed for this purpose. Metallographic tests and hardness measurements were performed to determine of influence of the welding process on the microstructure of the welded joint area, especially the heat affected zone (HAZ). The results obtained confirm the correctness of the assumptions made regarding the technology of manufacturing the furnace wall panels.

## 1. Introduction

Metal active gas (MAG) welding is a method of joining metal parts used in a variety of industries. Compared to the other methods [1], MAG produces highstrength welded joints at relatively low production costs.

In the MAG method, similar to other processes for manufacturing welded structures, the components to be joined are subjected to thermal cycles, i.e., heating and cooling, and phase transformations can occur in the structural material. Displacement of the heat source causes uneven heating and cooling of the component area around the weld, resulting in differential thermo-mechanical responses of the component. This results in stresses that, once exceeded, can cause deformation and strain in the joined components. This problem is particularly evident in complex, sophisticated welded structures [2,3].

Complete elimination of post-weld stress is difficult to achieve. However, efforts should be made to minimize their effect on the quality of the welded structure. Welding engineers have a number of solutions at their disposal. These include thermal and mechanical methods. Minimization of the influence of weld stress factors can also be achieved by proper selection of welding process parameters. These operations are often associated with increased financial outlay and extended production times [4,5].

With the development of computer technology, welding process simulation is increasingly used to optimize the production of welded structures. The use of the latest numerical simulation software for the welding process reduces the development time for the manufacturing technology of welded structures and the financial outlay [1,6].

Numerical simulations are used for the development of manufacturing processes for welded structures by means of various methods: MIG [2,4,7,8,9,10,11], MAG [7,8,9,12,13,14], TIG [1,7,15,16,17,18], NG-TIG [7], TIG-MIG [19], EBW [7], and WAAM [20,21].

Different types of software based on the Finite Element Method (FEM) are used for the numerical simulation of the welding process, in terms of welding stress and strain analysis. In works [1,5,13,14,18,22], the welding process is analyzed with the help of ABAQUS. In contrast, the authors of papers [3,8,10,12,15,16] used ANSYS for the numerical simulation of the welding process. Furthermore, in research on numerical simulations of the welding process, the SYSWELD program is used [4,7,17,23].

Numerical simulation of the welding process requires computers with high computing power. This is due to the large number of complex calculations involved. Therefore, various types of simplifications are used to reduce its implementation time. For example, to reduce the number of calculations in numerical simulations, axisymmetric elements are used, or 3D models are replaced by flat-transformed 2D elements [4,6,13,23].

The numerical simulation of a welding process should ensure that the results are as close as possible to the actual welding process (physical simulation). However, as the authors of papers [8,18] emphasize, this requires consideration of many variables describing the heat source, the welding method and its process parameters, the geometry of the welded joint, the welding technique, and the amount and type of restraint system (bracing).

The authors of works [5,13], analyzing the welding deformations and strains of large-scale welded structures, found a high correspondence between the results of the numerical simulation and those obtained under the conditions of the actual welding process (physical simulation).

The problem of welding stresses and deformations is particularly important in the fabrication of large welded structures consisting of a large number of welded joints made up of elements of varying thickness. In paper [12], numerical simulation in the ANSYS program was used to analyze the stress distribution arising in the welding process of a T-fillet joint of a Volvo truck chassis component. Some inconsistencies were found between the results of the stress distribution simulation and the results of residual stress measurements made in certain areas of the welded joint.

When welding large structures, the risk of weld distortion must be considered in the manufacturing technology. There are several methods used in the welding industry to minimize the distortion of welds. For example, the authors of paper [24] showed that the distribution and magnitude of weld distortion is strongly influenced by the method and sequence of weld layout. They analyzed six variants of weld placement in the welding process of a large plate with stiffening ribs made of the EH36 shipbuilding steel. They found that less weld distortion was achieved by placing the stiffener rib welds in the second variant, i.e., from the center to the edge of the plate.

The authors of paper [25] also found that the method and sequence of weld placement influences the amount of weld distortion produced during the welding process of large structures. Using numerical simulation, they analyzed the influence of the method and sequence of joint laying in the welding process of a vacuum vessel made of the 316LN austenitic steel, used in fusion processes.

High agreement between the results of numerical simulations of welding deformations and experimental results (physical simulation) in the case of welding of large constructions was demonstrated by the authors of paper [26]. They analyzed the welding process of a transport vehicle chassis beam made of the S235 steel using the MAG method. They also found a high correspondence between simulation results and experimental results in terms of temperature distribution in the welded joint area. They analyzed the influence of different methods for implementing the finite element method and found that the best results were obtained with the substructuring technique.

The aim of this work was to develop a MAG welding technology for a large construction furnace wall panel made of the S235JRC+N steel in terms of minimizing welding distortion, in accordance with the requirements of EN ISO 13920. The analysis of the welding technology was based on the results of numerical and physical simulation of the welding process, as well as visual inspections and measurements of the weld geometry. Metallographic tests and hardness measurements were performed to determine the effect of the welding process on the microstructure of the welded joint.

## 2. Materials and Methods

The subject of the study was an industrial furnace panel made of S235JRC+N steel with dimensions of 2000 mm × 4000 mm. The chemical composition of the S235JRC+N steel is shown in Table 1.

The furnace wall panel is a large welded structure made of sheet metal, flat bars, and a channel (Figure 1). The flat panel wall is made of 5 mm thick sheet metal, the outer frame of 200 mm × 80 mm channel sections with a wall thickness of 10 mm. The internal ribs are made from flat bars with a wall thickness of 6 mm and 80 mm × 80 mm angles with a wall thickness of 6 mm.

For the numerical simulation of the stress and strain distribution, a CAD and FEM model of the industrial furnace wall panel was created (Figure 2).

The FEM numerical model utilized a mesh made of second-order (2D) planar elements with center nodes. Using the CAD geometry, the center surfaces of the plates were determined and surfaces representing the welds were added. In the FE mesh, triangular or quadrilateral finite elements were used. In the weld areas, the mesh was compacted.

The numerical modeling of the production of the welded joints was carried out using the shrinkage method implemented in the SYSWELD program (ESI Group, Paris, France). This method consists of inducing shrinkage in the joint area with a value derived from the thermal expansion of the S235JRC+N steel. Sequential sections of the weld were defined by giving them a condition that induces shrinkage.

The welding sequence of the individual elements of the industrial furnace wall panel is shown in Figure 3. The welds were made in the back-step technique.

The MAG process is used in the production of furnace wall panels. The MAG welding process was performed using the technological parameters specified in the WPS (Welding Procedure Specification) developed in accordance with the WPQR document of the company manufacturing the industrial furnace wall panels.

The numerical simulation of the welding process of the industrial furnace wall panel was preformed assuming four models of restraint of the panel elements during the welding process (Figure 4).

All of the restraint system variants used the same values for the MAG welding process parameters and the same sequence of weld placement.

The boundary conditions of each restraint model took away degrees of freedom in selected directions in defined regions. These regions were consistent with the restraint model being analyzed.

Figure 5 shows model 1 of the distribution of restraint points.

A diagram of the distribution of restraints according to model 1 in the furnace wall panel construction during welding is shown in Figure 6.

For each restraint system model, the release of the panel from the fixture (restraint system) was realized by modification of the boundary conditions after completion of the welding process simulation. The post-weld restraint boundary conditions are shown in Figure 7.

The numerical simulation of the welding process of the furnace wall panel determined the distribution of the strain field and the deformation values at characteristic points on the furnace wall panel (Figure 8).

For the analysis of the numerical simulation results, deformations were specified in the direction of the X, Y, and Z axes. The values of these deformations were specified in order to be able to compare them with the results from the physical simulation.

The physical simulation consisted ofperforming the MAG welding process of the furnace wall panel. The welding process was performed using the 1st model of restraint. The view of the panel prepared for welding with the restraint system according to model 1 is shown in Figure 9.

Prior to the welding process, preliminary measurements were made. The measurements took into account the distances between characteristic points C1–C5, D1–D5, E1–E5, F1–F5 (direction of the X axis), and G1–G3 and H1–H3 (direction of the Y axis), as well as the values of relative height (in the direction of the Z axis) at characteristic measurement points. These measurements were carried out using a linear gauge (Stanley, Hartford, CT, USA) and a laser instrument (Bosch Power Tools, Warszawa, Poland). After the welding process, measurements were taken again.

Based on the deformation results in the numerical simulation and the results of measurements in the physical simulation of the welding process of the furnace wall panel, the tolerances for linear dimensions and tolerance for flatness were determined. These tolerances were calculated in accordance with EN ISO 13920.

To identify possible welding defects in the welded joints of the furnace wall panel, visual inspections were performed. These tests were carried out in accordance with the EN ISO 5817 standard. Measurements of the weld geometries (height and width of welds) were also taken.

Metallographic tests were carried out to reveal the microstructure of the welded joint area (weld, fusion zone, heat affected zone and parent material). Due to the large number of welds, one of the welds connecting the ribs made of flat bars to the plate (welds S.6–S.17 Figure 3c) was selected for metallographic examination.

In order to carry out metallographic examinations, it was necessary to prepare metallographic specimens (cutting, grinding, polishing) and to etch them properly, according to the commonly used methodology for steel specimens. Observations of the microstructure werecarried out using a NEOPHOT 2 optical microscope (Carl Zeiss, Jena, Germany).

Hardness measurements were carried out in accordance with the EN ISO 9015 standard, using the Vickers method at the HV5 load. On the basis of these measurements, the distribution of hardness in the weld, the heat affected zone, and the parent material weredetermined.

## 3. Results

### 3.1. Numerical and Physical Simulation

The distribution of the deformation fields of the industrial furnace wall panel in the direction of the X, Y, and Zaxes, resulting from the welding process with different restraint models, are shown in Figure 10, Figure 11 and Figure 12.

The deformation values at the characteristic measurement points (according to Figure 8), determined in the numerical simulation, are presented in Table 2, Table 3 and Table 4.

The largest displacements in the analyzed structure occurred in the direction of the Z axis, i.e., the direction perpendicular to the main surface of the furnace wall (Figure 12). Displacements in the −Z and +Z directions were found in individual areas of the welded element. The furnace wall panel was characterized by localized metal bulges between the ribs. There was also a general bulge in the entire panel.

The displacements in the X and Y directions were about fourtimes lower than the displacements in the Z direction.

For all restraint systems of the panel structure (models 1–4), the differences in deformation values were small, despite the fact that the number of stiffening points in model 1 was more than twice as small as in model 4.

Table 5 shows the deformation of the linear dimensions specified in the physical and numerical simulation of the welding process of the furnace wall panel.

The results of measuring linear dimension deformation indicate that the tolerance for line dimensions in physical simulation is from −2.00 to +1.00 mm. In the case of numerical simulation, the tolerance for line dimensions is from −1.77 to +2.18 mm. Considering the specifics, assumptions, and conditions for both simulations, it can be said that these tolerances are similar. However, deformations specified in the physical and numerical simulations differ in the plus and minus sign. This shows a different character of deformation, because the plus sign means the tensile stress, while the minus sign means the occurrence of compressive stresses.

According to EN ISO 13920, the tolerance for linear dimensions 2000–4000 mm in length (dimensions of the furnace wall panel) in class A should be ±4 mm. The values of tolerance for linear dimensions, specified in the physical and numerical simulations, are within this range.

The flatness tolerance was determined on the basis of the results of deformation measurements in the direction of the Z axis at characteristic points of the furnace wall panel. Table 6 presents the results of deformation measurements from the physical and numerical simulations of the welding process of the furnace wall panel. The mean absolute strain values obtained in the physical and numerical simulations are similar.

According to EN ISO 13920, in class F for the length range from 1000 to 2000 mm, the flatness tolerance is t = 4.5 mm. The obtained average absolute values of deformations at the measurement points from both simulations are within this range.

However, it can be noticed that the deformations at the characteristic points of the panel determined in the physical and numerical simulations are different in terms of value and sign (plus or minus). For numerical simulation, strain values have a plus sign. This proves that the predicted deformations arising in the panel welding process are the result of tensile stresses. In the case of physical simulation, the deformations have a plus or minus sign, which indicates that the process of welding the furnace wall panel generates tensile or compressive stresses.

As welding practice shows, there will always be stress in awelded structure. Compressive stresses occur in the weld as a result of solidification of the molten pool. On the other hand, tensile stresses occur in the heat-affected zone and the area of the welded material in this zone. The distribution of these stresses depends on many factors, including the welding method (heat source), the complexity of the welded structure, the thickness of the joined elements, and the welding process.

The methods used in welding processes differ in the source of heat. In the case of arc welding, the size, shape, and concentration of the heat source are determined by the values of the technological parameters of the welding method. This affects the course of welding thermal cycle, which results in the creation of a weld. This cycle includes heating and cooling of the welded joint area. The use of a wide electric arc with high power is associated with the risk of creating a wide heat-affected zone (HAZ) and a wide thermal impact zone on the parent material. In the case of the HAZ, a microstructure with low fracture toughness is formed in Fe-C alloys. In the areas further away from the weld, the thermal influence of the heat source may result in welding stresses due to cyclical heating and cooling. If the value of these stresses exceeds the value of the elastic limit, then plastic deformations, i.e., welding deformations, will occur. These deformations may also occur as a result of stresses arising in the adjacent welded joints (inhibition of weld shrinkage in the structures composed of a large number of welded joints). This is the case in the analyzed furnace wall panel; there are 38 welds.

Reducing the energy value of the arc and its width (high concentration) will result in a lower thermal impact on the material of the joined elements. This solution is used in the case of making welds of thick elements, which are joined with a multi-pass weld—the weld is created as a result of several passes of the electric arc. In addition, maintaining the appropriate temperature between the stitches reduces the adverse effect of the thermal cycle on the material of the joined elements. This reduces the risk of welding deformations.

The selection of technological parameters of the welding method, on the one hand, determines the energy value of the heat source, and on the other hand, the value of these parameters depends on the type and thickness of the elements. In the case of welded structures containing a large number of welds, these parameters also depend on the complexity of the structure and the method and sequence of arranging the welds.

The method and sequence of making the welds proposed in this paper are the result of research conducted as part of the optimization of the welding process of the furnace wall panel. Figure 3 shows the order in which the welds should be made and how they should be made with a step-back stitch. The choice of such a welding concept was the result of numerical simulations of deformations of many variants of the order and method of arranging the welds, which were carried out in the initial period of the optimization process.

The welding sequence is very important in the welding process. Welding in the correct order may result in partial relaxation of welding stresses. In the publication, the idea of the proposed welding sequences and methods of making welds of elements of the furnace wall panel (back-step technique—Figure 3) was to limit the possibility of generation of tensile and compressive stresses that could cause plastic deformations. Further minimization of welding deformations was obtained as a result of the use of a restraint system.

### 3.2. Visual Inspection and Measurement of Weld Geometry

The objective of visual testing (VT) was to assess the welded joints for the presence of welding defects. The results of the observations of the welded joints indicate the absence of welding defects. No excessive face overflow wasdetected. Analysis of the cross-section of the joint indicates the presence of full fusion. Small amounts of welding spatter were found on the joint surface.

Based on the measurement of the weld geometry, the average value of the weld face width over a length of 1 m was found to be 9.0 mm, while the average value of the weld face height over the analyzed length was 4.0 mm. These values areconsistent with the weld face width and height variation ranges adopted in the WPS manual It should be noted that the WPS technological instructions were developed in accordance with the existing WPQR procedures, because one of the criteria in the WPQR qualification process is the thickness of the joined elements. The thickness of these elements determines the thickness of the weld. For example, for butt welds made with full melting, the thickness of the weld is equal to the thickness of the joined elements. Depending on the thickness of the weld, the range of values of the other geometric parameters of the weld, i.e., the width and height of the weld face, is determined. The measured average values of the width and height of the weld face are within the ranges specified in the WPS and WPQR.

The microstructure of the welded joint was analyzed in individual areas of the joint (A–E), according to the diagram shown in Figure 13.

Area A (Figure 14) shows the microstructure of the parent material, i.e., S235JRC+N steel. This steel is structural steel in which the microstructure is obtained by hot rolling. The microstructure of S235JRC+N steel consists of ferrite and perlite (cementite and ferrite) distributed along the grain boundaries of the ferrite. The cementite in the perlite can take a lamellar or fibrous form.

Area B (Figure 15) shows the microstructure of the part of the HAZ furthest from the fusion boundary. In this part of the heat-affected zone, the effect of the heat source is finer gradation of the ferrite grains compared to the parent material.

The microstructure of area C, i.e., the fusion boundary, is shown in Figure 16. In this area, Widmanstätten ferrite (light) and bainite (dark) precipitates occur on the weld side. Ferrite (light) and bainite (dark) precipitates with different morphology are also present on the HAZ side.

The microstructure of the weld (area D) is shown in Figure 17. The weld contains Widmanstätten ferrite (light) and bainite (dark) precipitates.

Figure 18 shows the microstructure of area E, i.e., the location of the dual influence of the heat source. The microstructure of this area, as in Figure 15, is characterized by a greater degree of fragmentation of the ferrite grains compared to the parent material.

### 3.3. Hardness Measurements

The macrostructure of the welded joint area with the HV5 hardness measurement lines marked is shown in Figure 19.

The results of HV5 hardness measurements along measurement lines 1 and 2 are shown in Figure 20.

The results obtained show a similar distribution of the HV5 hardness on lines 1 and 2. The highest value of the HV5 hardness was found in the weld area, slightly lower in the HAZ and the lowest in the parent material.

The hardness value does not exceed 350HV5, which proves the lack of tendency of the welded joint material to form hard and brittle hardening products such as martensite.

## 4. Conclusions

As a conclusion of the research on the development of welding technology for a large industrial furnace wall panel, using numerical and physical simulation of the MAG welding process, it can be stated that:The use of different variants of the restraint system in the numerical simulation of the welding process of the industrial furnace wall panel gives a similar character of the deformation distribution, but with different values. The difference between the maximum mean strain in model 1 and the lowest mean strain in model 4 was found to be only 11%.The results of deformation measurements in the physical simulation of the furnace wall panel welding process differ from the results of the numerical simulation. These differences are visible in the values and the signs of the strains. This confirms the observations of other authors that numerical simulation of the welding process of alarge structures is difficult.Due to the requirements of EN ISO 13920 standard, all restraint models provide acceptable linear dimensional and flatness tolerances. Therefore, for the technology of welding a large wall panel of an industrial furnace, a variant of the constraint system according to model 1 is recommended.The results of microstructural tests and hardness measurements confirm the correctness of the developed technology, taking into account the recommended system of restraining the elements of the furnace wall panel during the welding process and the proposed sequence and method of welding.

## Figures and Tables

**Figure 1 materials-16-02779-f001:**
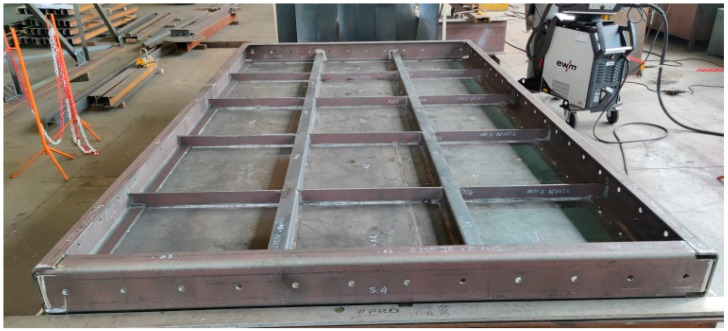
Furnace wall panel elements connected by spot welds.

**Figure 2 materials-16-02779-f002:**
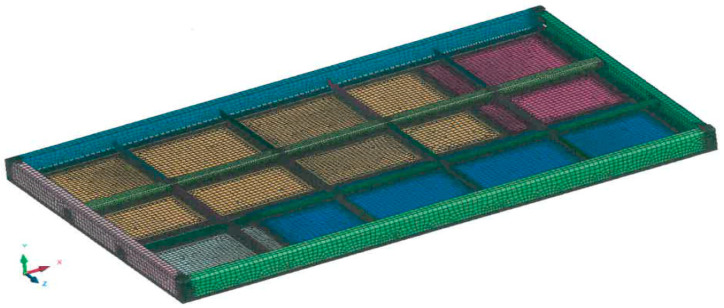
CAD and FEM model of the industrial furnace wall panel.

**Figure 3 materials-16-02779-f003:**
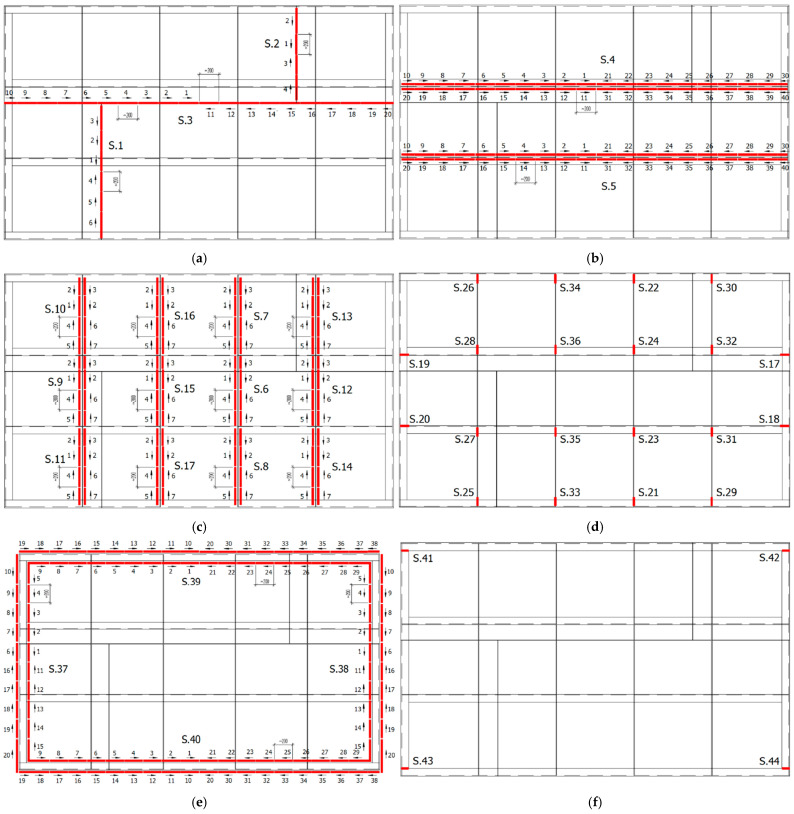
Sequence of welds in the numerical simulation of the welding process of the industrial furnace wall panel; (**a**) welds S.1–S.3 joining metal sheets (stage 1), (**b**) welds S.4 and S.5 joining ribs made of angles to the metal sheets (stage 2), (**c**) welds S.6–S.17 joining the ribs made of flat bars to the metal sheets (stage 3), (**d**) welds S.18–S.36 joining the ribs made of flat bars to the channel sections of the frame and ribs made of angles (stage 4), (**e**) welds S.37–S.40 joining the channel sections of the outer frame to the metal sheets (stage 5), and (**f**) welds S.41–S.44 joining the channel sections of the outer frame (stage 6).

**Figure 4 materials-16-02779-f004:**
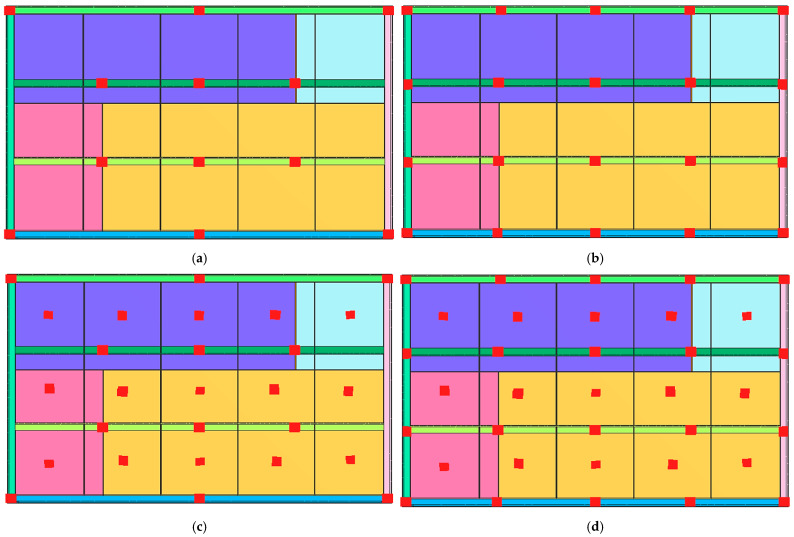
Diagram of the distribution of restraint points in four variants of the numerical simulation of the welding process of the industrial furnace wall panel; (**a**) model 1, (**b**) model 2, (**c**) model 3, and (**d**) model 4.

**Figure 5 materials-16-02779-f005:**
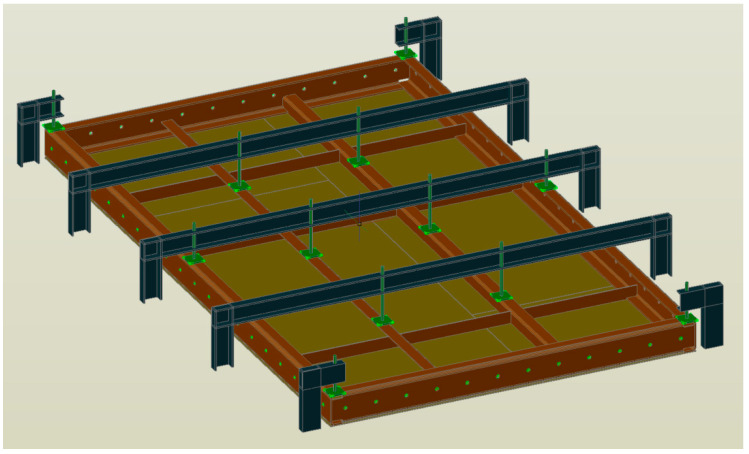
Diagram of the distribution of restraint points in model 1 (Figure 3a) of the welding process of the industrial furnace wall panel.

**Figure 6 materials-16-02779-f006:**
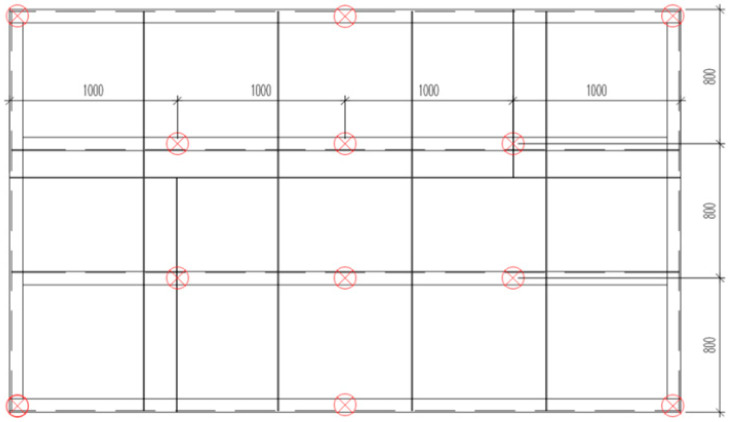
Diagram of the distribution of restraints in the furnace wall panel structure during welding.

**Figure 7 materials-16-02779-f007:**
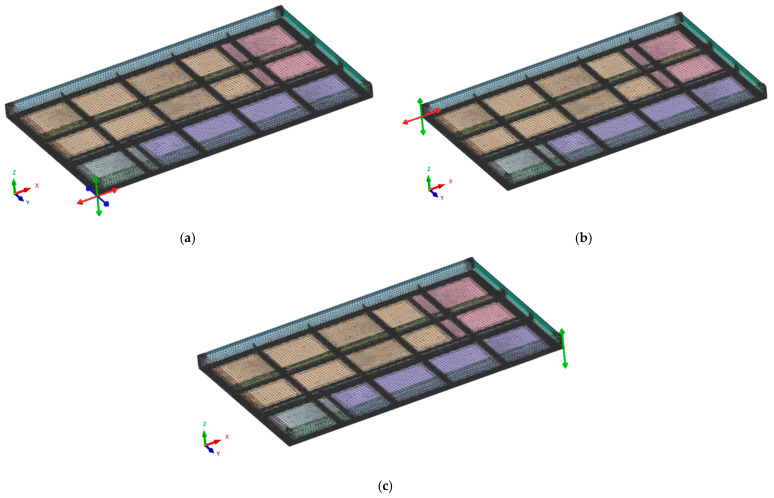
Diagrams of restraint boundary conditions after panel welding; (**a**) no node displacement in the X, Y, and Z directions, (**b**) no node displacement in the X and Y directions, and (**c**) no node displacement in the Y direction.

**Figure 8 materials-16-02779-f008:**
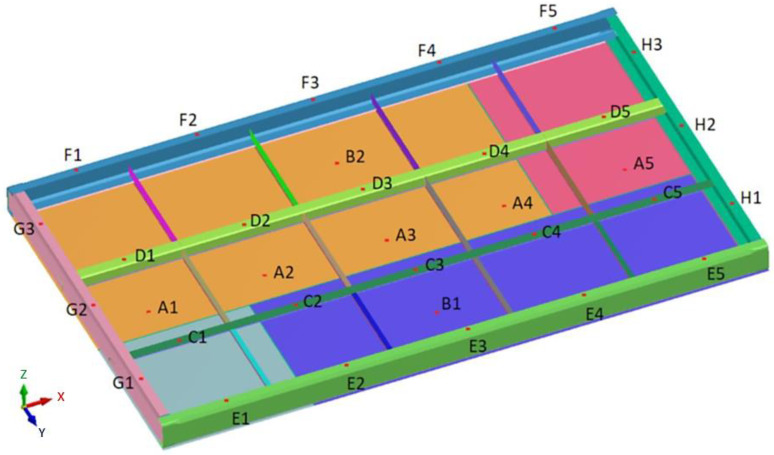
Distribution of selected measurement points where deformation values were determined in the numerical simulation of the furnace wall panel welding process.

**Figure 9 materials-16-02779-f009:**
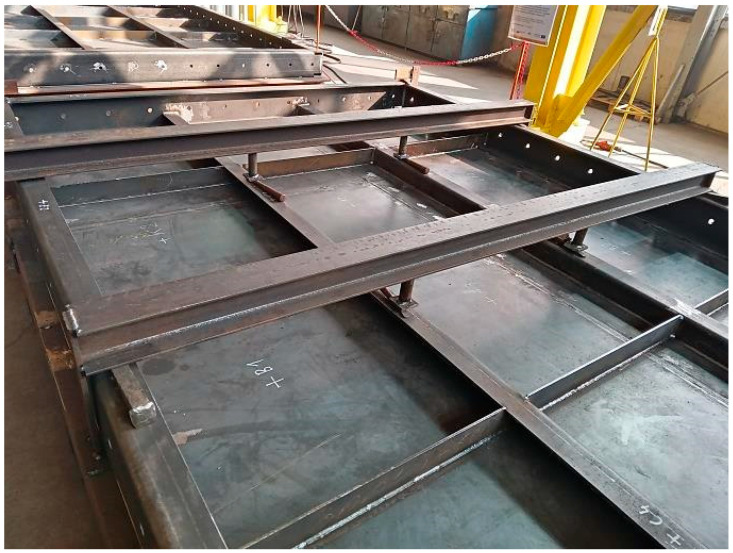
View of the furnace wall panel elements connected by spot welds to the restraint system according to model 1.

**Figure 10 materials-16-02779-f010:**
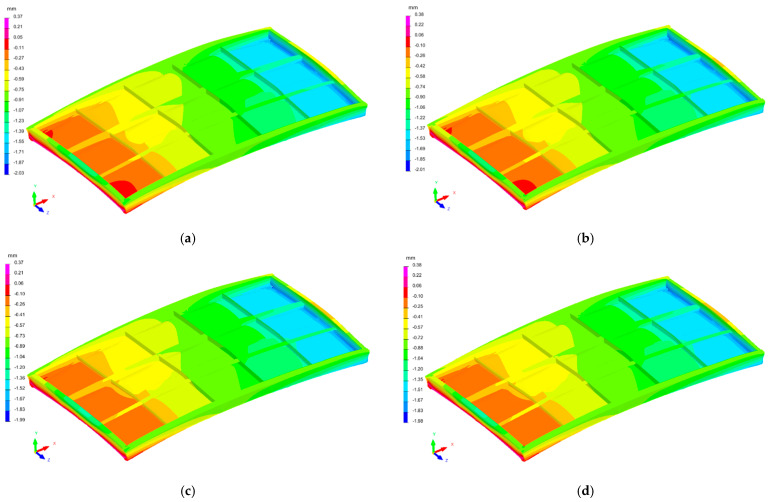
Distribution of the strain field of the industrial furnace wall panel in the X direction; (**a**) 1st restraint model, (**b**) 2nd restraint model, (**c**) 3rd restraint model, and (**d**) 4th restraint model.

**Figure 11 materials-16-02779-f011:**
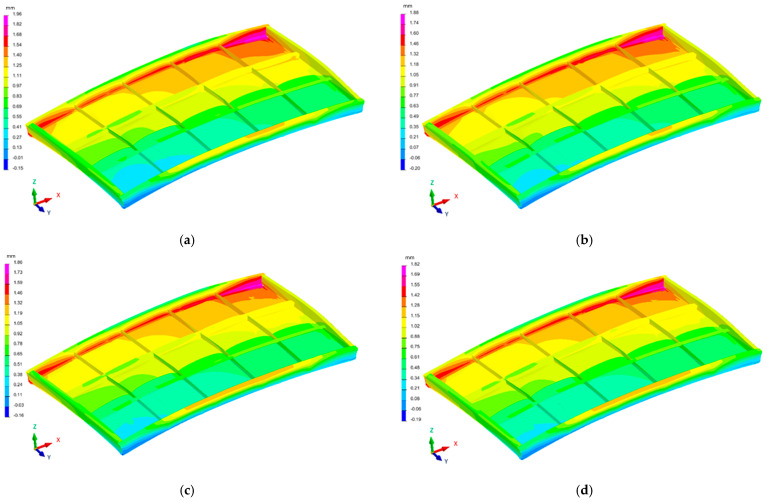
Distribution of the strain field of the industrial furnace wall panel in the Y direction; (**a**) 1st restraint model, (**b**) 2nd restraint model, (**c**) 3rd restraint model, and (**d**) 4th restraint model.

**Figure 12 materials-16-02779-f012:**
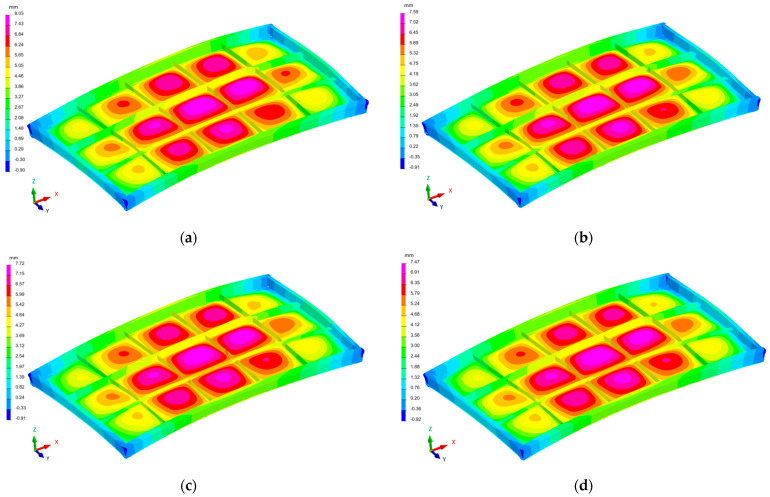
Distribution of the strain field of the industrial furnace wall panel in the Z direction; (**a**) 1st restraint model, (**b**) 2nd restraint model, (**c**) 3rd restraint model, and (**d**) 4th restraint model.

**Figure 13 materials-16-02779-f013:**
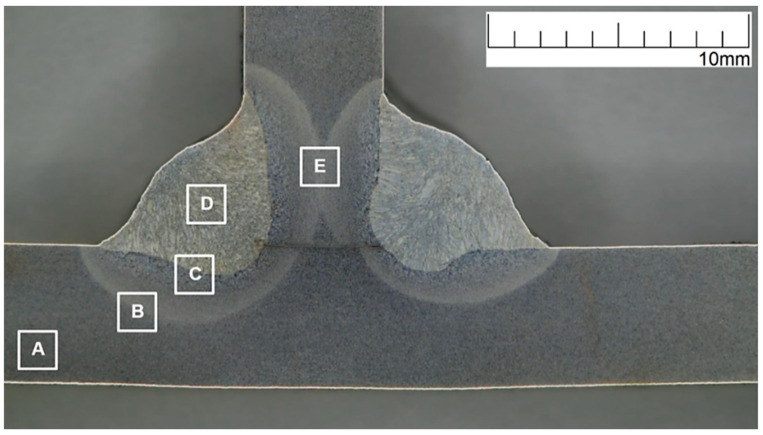
Macrostructure of the test joint with marked areas of the microstructure test: A—parent material, B—heat-affected zone (HAZ), C—weld and heat-affected zone boundary, D—weld, and E—heat-affected zone (HAZ).

**Figure 14 materials-16-02779-f014:**
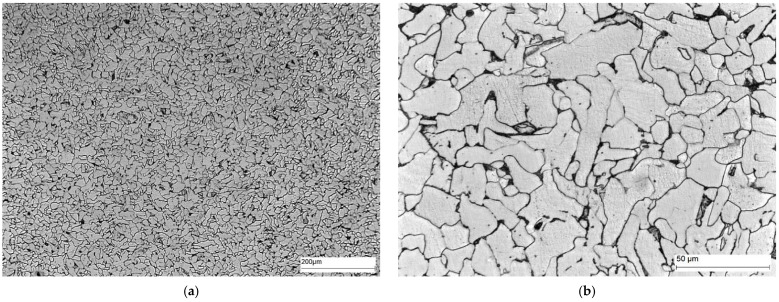
Microstructure of area A of the welded joint—parent material: (**a**) 100× magnification, and (**b**) 500× magnification. Ferrite—white precipitates, and perlite—dark precipitates at the ferrite grain boundaries.

**Figure 15 materials-16-02779-f015:**
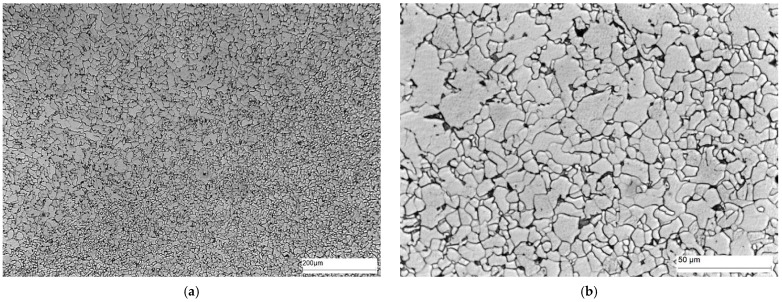
Microstructure of area B of welded joint—HAZ: (**a**) magnification 100×, and (**b**) magnification 500×. Ferrite—white precipitates, and perlite—dark precipitates at the ferrite grain boundary.

**Figure 16 materials-16-02779-f016:**
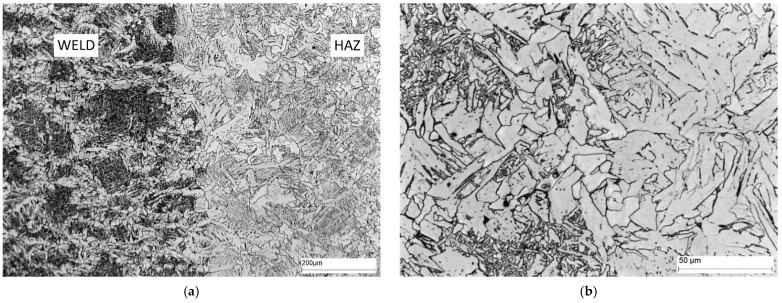
Microstructure of area C—fusion boundary: (**a**) 100× magnification, and (**b**) 500× magnification. Widmanstätten ferrite—white separations, hardening products, mainly bainite—dark separations.

**Figure 17 materials-16-02779-f017:**
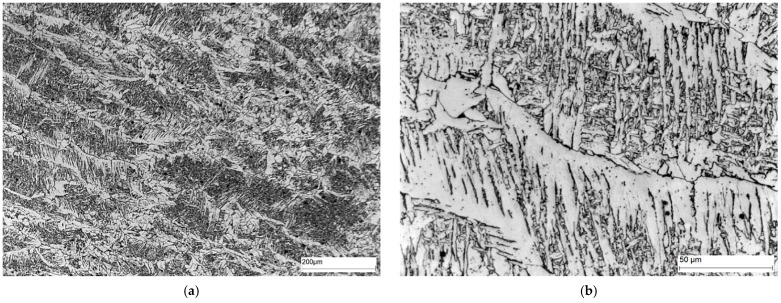
Microstructure of area D—weld: (**a**) 100× magnification, and (**b**) 500× magnification. Widmanstätten ferrite—white precipitates with different morphology, quenching products, mainly bainite—dark precipitates.

**Figure 18 materials-16-02779-f018:**
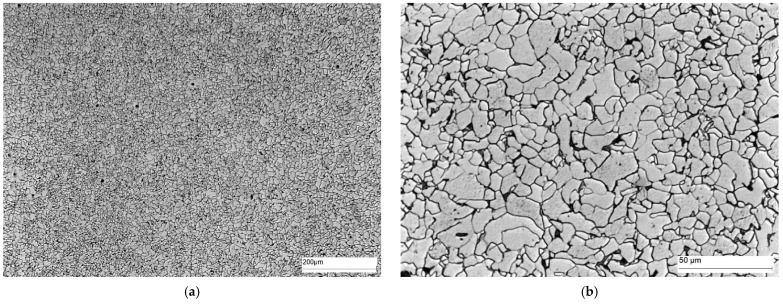
Microstructure of area E of welded joint—SWC: (**a**) magnification 100×, and (**b**) magnification 500. Ferrite—white precipitates, and perlite—dark precipitates at grain boundaries.

**Figure 19 materials-16-02779-f019:**
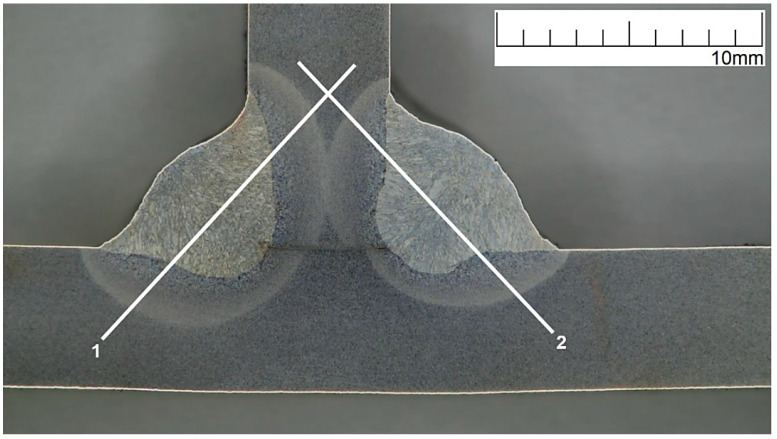
Macrostructure of welded joint with HV5 hardness measurement lines marked.

**Figure 20 materials-16-02779-f020:**
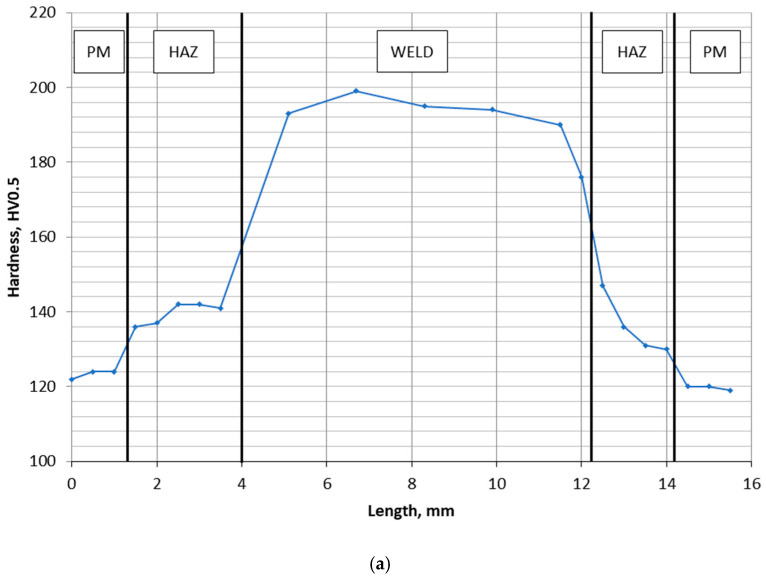
HV5 hardness distribution in the welded joint area; (**a**) measurement line 1, and (**b**) measurement line 2. PM—parent material, HAZ—heat-affected zone.

**Table 1 materials-16-02779-t001:** Chemical composition of steel S235JRC+N (from metallurgical certificate) in wt.%.

Fe	C	Mn	P	S	Cu	N
balance	0.14	0.48	0.013	0.007	0.014	0.003

**Table 2 materials-16-02779-t002:** Results, average value, and sum of displacements in the X direction at characteristic measurement points, obtained in the numerical simulation of the welding process of the furnace wall panel.

**Model**	**A1**	**A2**	**A3**	**A4**	**A5**	**B1**	**B3**	**C1**	**C2**	**C3**	**C4**	**C5**	**D1**	**D2**	**D3**	**D4**	**D5**	**E1**
1	−0.19	−0.49	−0.79	−1.02	−1.50	−0.89	−0.71	−0.41	−0.66	−0.85	−1.03	−1.24	−0.37	−0.59	−0.78	−1.00	−1.28	−0.88
2	−0.19	−0.48	−0.77	−1.00	−1.48	−0.87	−0.71	−0.41	−0.64	−0.83	−1.01	−1.20	−0.37	−0.58	−0.77	−0.98	−1.24	−0.85
3	−0.22	−0.50	−0.77	−1.00	−1.43	−0.86	−0.71	−0.42	−0.64	−0.83	−1.00	−1.18	−0.39	−0.58	−0.76	−0.97	−1.21	−0.85
4	−0.21	−0.49	−0.77	−0.99	−1.42	−0.85	−0.71	−0.42	−0.63	−0.82	−0.99	−1.16	−0.39	−0.58	−0.76	−0.97	−1.20	−0.84
**Model**	**E2**	**E3**	**E4**	**E5**	**F1**	**F2**	**F3**	**F4**	**F5**	**G1**	**G2**	**G3**	**H1**	**H2**	**H3**			
1	−0.85	−0.86	−0.88	−0.89	−0.75	−0.77	−0.79	−0.79	−0.77	−0.89	−1.15	−0.72	−0.81	−0.46	−0.80			
2	−0.83	−0.84	−0.86	−0.87	−0.72	−0.74	−0.76	−0.76	−0.75	−0.84	−1.15	−0.69	−0.81	−0.41	−0.77			
3	−0.83	−0.83	−0.85	−0.86	−0.73	−0.74	−0.76	−0.76	−0.75	−0.89	−1.19	−0.73	−0.76	−0.37	−0.74			
4	−0.82	−0.83	−0.85	−0.86	−0.71	−0.73	−0.75	−0.75	−0.74	−0.87	−1.20	−0.72	−0.76	−0.33	−0.72			

**Table 3 materials-16-02779-t003:** Results, average value, and sum of displacements in the Y direction at characteristic measurement points, obtained in the numerical simulation of the welding process of the furnace wall panel.

**Model**	**A1**	**A2**	**A3**	**A4**	**A5**	**B1**	**B3**	**C1**	**C2**	**C3**	**C4**	**C5**	**D1**	**D2**	**D3**	**D4**	**D5**	**E1**
1	0.95	0.98	1.05	1.11	1.15	0.51	1.29	0.69	0.72	0.82	0.94	0.91	0.98	0.93	1.04	1.14	1.17	0.66
2	0.91	0.93	0.99	1.05	1.08	0.45	1.23	0.65	0.65	0.75	0.86	0.84	0.92	0.89	0.98	1.08	1.08	0.57
3	0.89	0.92	0.98	1.04	1.07	0.50	1.20	0.65	0.67	0.77	0.88	0.85	0.91	0.89	0.98	1.08	1.09	0.64
4	0.86	0.89	0.95	1.00	1.03	0.47	1.17	0.63	0.64	0.73	0.83	0.81	0.88	0.86	0.95	1.04	1.03	0.59
**Model**	**E2**	**E3**	**E4**	**E5**	**F1**	**F2**	**F3**	**F4**	**F5**	**G1**	**G2**	**G3**	**H1**	**H2**	**H3**			
1	1.21	1.25	1.24	0.88	1.08	0.77	0.74	0.68	1.23	0.83	0.82	0.82	1.09	1.09	1.09			
2	1.11	1.17	1.12	0.75	1.06	0.74	0.67	0.65	1.15	0.76	0.77	0.79	0.97	0.98	1.00			
3	1.22	1.26	1.22	0.81	1.01	0.66	0.62	0.56	1.12	0.78	0.78	0.78	1.00	1.00	1.00			
4	1.16	1.21	1.16	0.75	0.99	0.64	0.58	0.55	1.07	0.74	0.75	0.77	0.93	0.94	0.95			

**Table 4 materials-16-02779-t004:** Results, average value, and sum of displacements in the Z direction at characteristic measurement points, obtained in the numerical simulation of the welding process of the furnace wall panel.

**Model**	**A1**	**A2**	**A3**	**A4**	**A5**	**B1**	**B3**	**C1**	**C2**	**C3**	**C4**	**C5**	**D1**	**D2**	**D3**	**D4**	**D5**	**E1**
1	5.59	7.43	8.01	7.94	6.14	7.58	7.36	2.82	4.68	4.98	4.56	2.94	2.40	3.83	4.55	4.67	2.86	1.56
2	5.23	7.05	7.57	7.48	5.64	7.27	6.94	2.44	4.32	4.58	4.15	2.46	2.03	3.44	4.09	4.18	2.28	1.42
3	5.25	7.13	7.70	7.56	5.68	7.28	6.93	2.81	4.72	5.00	4.57	2.87	2.38	3.85	4.53	4.60	2.68	1.50
4	5.05	6.90	7.45	7.29	5.41	7.10	6.69	2.57	4.51	4.78	4.33	2.57	2.15	3.63	4.27	4.32	2.34	1.43
**Model**	**E2**	**E3**	**E4**	**E5**	**F1**	**F2**	**F3**	**F4**	**F5**	**G1**	**G2**	**G3**	**H1**	**H2**	**H3**			
1	3.59	3.80	3.02	1.14	1.21	3.08	3.91	3.78	1.85	0.95	1.78	0.70	1.01	2.11	1.28			
2	3.32	3.58	2.82	1.05	1.09	2.78	3.51	3.23	1.39	0.65	1.34	0.46	0.65	1.47	0.72			
3	3.54	3.80	3.01	1.10	1.18	3.02	3.81	3.57	1.58	0.87	1.67	0.64	0.90	1.87	1.02			
4	3.38	3.67	2.89	1.06	1.11	2.85	3.58	3.26	1.33	0.68	1.39	0.50	0.67	1.49	0.69			

**Table 5 materials-16-02779-t005:** Deformation of the linear dimensions specified in the physical and numerical simulation of the welding process of the industrial furnace wall panel.

Distance	Deformation, mm				
	Physical Simulation	Numerical Simulation			
		Model 1	Model 2	Model 3	Model 4
C1–C5	−2.00	−1.65	−1.61	−1.60	−1.58
D1–D5	−2.00	−1.65	−1.61	−1.60	−1.59
E1–E5	0.00	−1.77	−1.72	−1.71	−1.70
F1–F5	+1.00	−1.52	−1.47	−1.48	−1.45
G1–G3	0.00	+1.65	+1.55	+1.56	+1.52
H1–H3	−2.00	+2.18	+1.97	+2.00	+1.88

**Table 6 materials-16-02779-t006:** Deformations in the Z direction at the measurement points of the furnace wall panel (physical and numerical simulation).

Point	Deformation in the Direction of the Z axis, mm				
	Physical Simulation	Numerical Simulation			
		Model 1	Model 2	Model 3	Model 4
A1	+4	+5.59	+5.23	+5.25	+5.05
A2	+6	+7.43	+7.05	+7.13	+6.90
A3	+7	+8.01	+7.57	+7.70	+7.45
A4	+5	+7.94	+7.48	+7.56	+7.29
A5	+13	+6.14	+5.64	+5.68	+5.41
B1	+6	+7.58	+7.27	+7.28	+7.10
B2	+6	+7.36	+6.94	+6.93	+6.69
C1	−2	+2.82	+2.44	+2.81	+2.57
C2	−2	+4.68	+4.32	+4.72	+4.51
C3	−1	+4.98	+4.58	+5.00	+4.78
C4	−8	+4.56	+4.15	+4.57	+4.33
C5	−3	+2.94	+2.46	+2.87	+2.57
D1	−3	+2.40	+2.03	+2.38	+2.15
D2	−1	+3.83	+3.44	+3.85	+3.63
D3	+1	+4.55	+4.09	+4.53	+4.27
D4	+1	+4.67	+4.18	+4.60	+4.32
D5	−1	+2.86	+2.28	+2.68	+2.34
E1	+5	+1.56	+1.42	+1.50	+1.43
E2	+4	+3.59	+3.32	+3.54	+3.38
E3	+1	+3.80	+3.58	+3.80	+3.67
E4	0	+3.02	+2.82	+3.01	+2.89
E5	−2	+1.14	+1.05	+1.10	+1.06
F1	−4	+1.21	+1.09	+1.18	+1.11
F2	+1	+3.08	+2.78	+3.02	+2.85
F3	+1	+3.91	+3.51	+3.81	+3.58
F4	−1	+3.78	+3.23	+3.57	+3.26
F5	−4	+1.85	+1.39	+1.58	+1.33
G1	−1	+0.95	+0.65	+0.87	+0.68
G2	−3	+1.78	+1.34	+1.67	+1.39
G3	−3	+0.70	v0.46	+0.64	+0.50
H1	−3	+1.01	+0.65	+0.90	+0.67
H2	0	+2.11	+1.47	+1.87	+1.49
H3	5	+1.28	+0.72	+1.02	+0.69
Average of absolute values	**3.27**	**3.73**	**3.35**	**3.59**	**3.37**

## Data Availability

Data is contained within the article.

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
