# Peer review of "Numerical and Physical Simulation of MAG Welding of Large S235JRC+N Steel Industrial Furnace Wall Panel"

_materials, 2023, doi:10.3390/ma16072779_

Round 1

Reviewer 1 Report

The paper entitled “Numerical and physical simulation of MAG welding of a large industrial furnace wall panel made of S235JRC+N steel" presents the results of a study on the development of a MAG welding technology for an industrial furnace component made of steel S235JRC+N. From my point of view, the topic is of great interest and the quality is good. But in general:

·        In the abstract when the author mention the novelty of the paper it is not clear explanation.

·        References to figures in capitals (e.g. (fig. 1).

·        In the introduction to include the application of finite element modelling in welding derived processes such as WAAM, I believe it can add a focus to the application of this type of work to other processes. As examples:

o   https://doi.org/10.3390/met11050678

o   https://doi.org/10.1016/j.commatsci.2011.06.023

·        Figure 10-12 include units in the scale.

·        The truth is that the article is very interesting, but the exposition of the results is very expository and not very argumentative.

·        The quality of Fig. 20 can be improved.

The paper has important work in the modelling part but the experimental verification could be more adequate but I think it has value for publication

Reviewer 2 Report

1. The results obtained by your model 1 in the two kinds of simulation are quite different, and the reliability of the numerical simulation results has not been proved. Please response it. 

2. The relationship between welding induced deformation and weld quality should be identified.

3.  In the result part, the evaluation method for the quality of welded joints is not comprehensive and scientific enough, which does not take into account the strength of welded joints. Please explain it. 

Reviewer 3 Report

This review article the stress-strain and joint quality problems of steel S235JRC+N of MIG welding. It plays a guiding role in the numerical simulation of welding of large structural parts. So this manuscript can be accepted and published under major revision. The modifications are shown below:

1.There are too many paragraphs divided into articles. Please simplify the language and integrate long paragraphs.

2.The article lacks an introduction to the models and principles of numerical simulation.

3.In the part of 3.2, there are too little content. Could the author add some pictures or content.

4.The aesthetics of the accompanying pictures need to be improved.

5.The quality of English needs improving.

6.Please add the following the paper in your references.

RAO Lei, ZHAO Jian-hua, ZHAO Zhan-xi, DING Gang, GENG Mao-peng, Macro- and Microstructure Evolution of 5CrNiMo Steel Ingots during Electroslag Remelting Process, Journal of Iron and Steel Research, International, Volume 21, Issue 7, 2014, Pages 644-652.

A-rong, Lin ZHAO, Chuan PAN, Zhi-ling TIAN, Influence of Ti on Weld Microstructure and Mechanical Properties in Large Heat Input Welding of High Strength Low Alloy Steels, Journal of Iron and Steel Research, International, Volume 22, Issue 5, 2015, Pages 431-437.

Reviewer 4 Report

Manuscript ID: Materials-2286138 entitled "Numerical and physical simulation of MAG welding of a large industrial furnace wall panel made of S235JRC+N steel" for journal of "Materials" has been reviewed.

- This article is comprehensive, logically organized and contains valuable information. However, there are few things need to be corrected and included in the manuscript for better understanding of carried research work to the readers.

+1- The novelty of the study should be mentioned a little more in the introduction section.

+2- More references should be added to the introduction section. (especially about different studies)

+3- More information on tests should be given in the "Materials and Methods " section. (humidity, ambient temperature, etc.).

+4- …. The flatness tolerance was determined on the basis of the results of deformation measurements in the direction of the Z axis at characteristic points of the furnace wall panel. Table 6 presents the results of deformation measurements from the physical and numerical simulation of the welding process of the furnace wall panel. The mean absolute strain values obtained in the physical and numerical simulations are similar. According to EN ISO 13920, in class F for the length range from 1000 to 2000 mm, the flatness tolerance is t = 4.5 mm. The obtained average absolute values of deformations at the measurement points from both simulations are within this range. However, it can be noticed that the deformations at the characteristic points of the panel determined in the physical and numerical simulation are different in terms of value and sign (plus or minus). For numerical simulation, strain values have a plus sign. This proves that the predicted deformations arising in the panel welding process are the result of tensile stresses. In the case of physical simulation, the deformations have a plus or minus sign, which indicates that the process of welding the furnace wall panel generates tensile or compressive stresses. As welding practice shows, stresses will always arise in the welded structure. Compressive stresses arise in the weld and are the result of solidification of the liquid metal pool. On the other hand, tensile stresses arise in the heat-affected zone and the area of the welded material at this zone. The distribution of these stresses depends on many factors, including the welding method (heat source), the complexity of the welded structure, the thickness of the joined elements and the welding process…..(please explain more, why?)

+5- … The objective of the visual testing (VT) was to assess the welded joints for the presence of weld defects. The results of the observations of the welded joints indicate the absence of welding defects. No excessive face overflow was found. Analysis of the cross-section of the joint indicates the presence of full fusion. Small amounts of welding spatter were found on the joint surface. Based on the measurement of the weld geometry, the average value of the width of the weld face over a length of 1 m was found to be 9.0 mm, while the average value of the height of the weld face over the analyzed length was 4.0 mm. These values are in accordance with the weld face width and height variation ranges adopted in the WPS manual…. (please explain in more detail.)

6- The resolution of Figures 4 (and magnify), 8 and 9 should be increased. The thickness of the lines in the graphic (fig. 9) should be increased. Colours are not clear.

+7- More literature studies should be added to the introduction and other sections (DOIs given below).

DOI-1  https://doi.org/10.26701/ems.989945   (info about different studies)

DOI-2  https://doi.org/10.1007/s13369-021-06243-w    (info about different studies)

DOI-3  https://doi.org/10.35193/bseufbd.1075980    (info for tests… humidity, ambient temp., etc.)

-----------------------------------------------------------------------

* It will be ready for publication after the specified corrections.

** I want to see article after the revision.

-----------------------------------------------------------------------

Congratulations to the authors.

I wish the authors success in their future academic studies.

Kind regards.

Round 2

Reviewer 1 Report

The paper is OK now

Reviewer 4 Report

Manuscript ID: Materials-2286138 entitled (updated by author) “Numerical and physical simulation of MAG welding of large S235JRC+N steel industrial furnace wall panel" for journal of "Materials" has been reviewed.

----------------------------------------------------------------

Abstract (updated by author): This paper presents the results of a study on the development of a MAG welding technology for an industrial furnace component made of steel S235JRC+N with respect to the minimization of welding deformation. A numerical simulation of the welding process was performed in the first phase of the research. The numerical simulation was carried out with the SYSWELD software. For the numerical simulation of the welding process, the FEM method was used. In the simulation, four variants of restraint of the industrial furnace wall panel elements during the execution of the welding process were investigated. They differed in the number of restraints (model 1-4). It was found that the difference between the maximum mean strain in model 1 and the lowest mean strain in model 4 was only 11%. A physical simulation of the welding process was then performed with a restraint variant according to model 1. The displacement results obtained from the physical simulation of the welding process were compared with the displacement results from the numerical simulation. Discrepancies between numerical and physical simulation displacement values were found. The quality of selected welded joints was also evaluated. Visual testing (VT) and measurements of weld geometries were performed fot this purpose. Metallographic tests and hardness measurements were performed to determine of influence of the welding process on the microstructure of the welded joint area, especially the heat affected zone (HAZ). The results obtained confirm the correctness of the assumptions made regarding the technology of manufacturing the furnace wall panels.

-----------------------------------------------------------------------------

-

Title of manuscript- Suitable with contex

-Abstract clearly presents objects methods and results.

-Scientific methods are adequately used.

-Terminology is adequate.

-Results are clearly presented .

-Conclusions are logically derived from the data presented.

-Key words are adequate.

-References are appropriate.

-

The authors have made the necessary corrections.

The article is suitable for publication.

Decision- Accept

-----------------------------------------------------------------

Congratulations to the authors.

I wish the authors success in their future academic studies.

Kind regards.